# Emotional Contagion and Emotional Mimicry in Individuals with Schizophrenia: A Systematic Review

**DOI:** 10.3390/jcm13175296

**Published:** 2024-09-06

**Authors:** Mathilde Parisi, Ludovic Marin, Tifenn Fauviaux, Emilie Aigoin, Stéphane Raffard

**Affiliations:** 1EuroMov Digital Health in Motion, University of Montpellier, IMT Mines Ales Montpellier, 700 Avenue du Pic Saint Loup, 34090 Montpellier, France; mathilde.parisi@umontpellier.fr (M.P.); ludovic.marin@umontpellier.fr (L.M.); tifenn.fauviaux.35@gmail.com (T.F.);; 2Faculty of Psychology, Univ Paul Valéry Montpellier 3, Univ. Montpellier, Laboratory EPSYLON EA 4556, 34090 Montpellier, France; 3University Department of Adult Psychiatry, CHU Montpellier, 34000 Montpellier, France

**Keywords:** psychosis, emotion, social skills, imitation, empathy

## Abstract

**Background:** Individuals with schizophrenia often exhibit social interaction deficits, which can affect their ability to engage effectively with others. Emotional processes, such as emotional contagion (the transfer of emotion between individuals) and emotional mimicry (the imitation of emotional expressions), are crucial for enhancing the quality of social interactions. **Methods:** We conducted a PubMed, Web of Science, and PsycInfo database search. The inclusion and exclusion criteria were established based on the definitions of emotional contagion and emotional mimicry, rather than relying on specific terminology from various research fields. Forty-two studies were included in the review, including six emotional mimicry studies and thirty-six emotional contagion studies. **Results:** The current findings suggest decreased or inappropriate emotional mimicry in individuals with schizophrenia. Relating to emotional contagion, the results showed altered brain and psychophysiological activity in individuals with schizophrenia, whereas the self-reported measures indicated no difference between the groups. The relationships between emotional contagion, emotional mimicry, and psychotic symptom severity showed variability across the studies, whereas no associations between antipsychotic dosage and either emotional mimicry or emotional contagion were found. **Discussion:** This review highlights the need to further evaluate and train emotional contagion and emotional mimicry in individuals with schizophrenia because these processess influence social interaction quality. Clinical implications and guidelines for future studies are discussed.

## 1. Introduction 

Schizophrenia is a psychiatric diagnosis characterized by a highly heterogeneous clinical presentation. Different types of symptoms are identified: positive (e.g., hallucinatory experiences), negative (e.g., blunted affect), and cognitive (e.g., impaired working memory). In addition, deficits in social functioning are a core feature of schizophrenia [1,2]. 

Because emotions serve social functions, such as allowing people to form and maintain social relationships [3], many researchers started investigating emotional processes in individuals with schizophrenia. Different emotional alterations were identified, as follows: reduced performance in emotion perception and recognition tasks, diminished emotional expressivity, decreased emotional intelligence, and increased feelings of anhedonia [4,5]. Two other emotional processes, emotional mimicry (i.e., the imitation of emotional expression) [6] and emotional contagion (i.e., the transfer of emotion between individuals) [7,8], are particularly interesting to study in the context of schizophrenia as they are also closely linked to social outcomes. For instance, emotional contagion is a precursor of empathy, which was shown to be altered in individuals with schizophrenia [9,10]. In addition, emotional mimicry acts as a social regulator that is associated with reciprocal liking and social interaction quality [11,12,13].

Motional contagion was first defined as the “tendency to automatically mimic and synchronize facial expressions, vocalizations, postures and movements with those of another person and, consequently, converge emotionally” (p. 5) [8]. According to this definition, emotional contagion is a three-step process: the first step is the mimicry of an emotional expression, the second is afferent feedback, and the third is emotional convergence. This definition of emotional contagion is referred to as primitive emotional contagion due to its description as an automatic and bottom-up process. Two recent literature reviews proposed that emotional contagion can be measured through a participant’s self-reported emotional states, self-reported susceptibility, behavioral expressions, and psychophysiological reactions to another person’s emotional expression [7,14].

However, Hatfield and colleagues’ definition of emotional contagion was contested for different reasons. First, some studies were not able to demonstrate the causal link between mimicry and contagion [15,16,17]. In addition, emotional mimicry is not an automatic reaction; rather, it performs social functions, such as the mimicker showing their comprehension of their counterpart’s emotion [6]. As a result, although certain studies have demonstrated the connection between or co-occurrence of emotional contagion and emotional mimicry [18,19], the limitations of Hatfield’s definition have led to the distinction between the two phenomena [6]. In addition, in the distinction between emotional contagion and emotional mimicry, it has been noted that “emotional contagion refers to a feeling state whereas mimicry refers to an overt behavior”, highlighting the different variables of interest of the two phenomena [11].

Consequently, for this review, we selected articles based on the following definitions. Emotional contagion refers to the transfer of an emotional state from one individual to another through emotional expressions, leading to emotional convergence. It is typically measured by participants’ self-reported emotions, brain activity, or psychophysiological responses to another individual’s emotional expressions, reflecting their own emotional experiences. Conversely, emotional mimicry refers to the imitation of emotional expressions, resulting in corresponding behaviors. It is measured by observing the participant’s emotional expressions in response to another individual’s emotional expression.

To our knowledge, no systematic review on emotional contagion and emotional mimicry in schizophrenia has been conducted to date. In addition, there are two major difficulties in searching for studies about emotional contagion and emotional mimicry in individuals with schizophrenia highlighting the need for a systematic review. First, the variety of fields interested in measuring emotional contagion has led to the usage of different terminologies. For example, neurosciences include emotional contagion in the broader term, “emotional processing”, which also refers to other mechanisms, like emotional regulation. In Psychology, many expressions are used, such as interpersonal emotional transfer [20] or social induction of affect [21]. Second, the terminology discrepancies make it difficult to ensure whether a study is measuring emotional contagion and emotional mimicry. For instance, emotional contagion and emotional mimicry can be confused with emotional induction or the measurement of blunted affect. Thus, clear descriptions of experimental procedures allowing the measurement of emotional contagion and emotional mimicry are needed to report on studies in individuals with schizophrenia. Finally, because of the variety of methodologies used (e.g., self-reported emotions, brain activity, psychophysiological reactions), a systematic review enables us to confront the conclusion across the fields. 

Consequently, this systematic review aimed to report emotional mimicry and emotional contagion measures in individuals with schizophrenia to emphasize possible implications for individuals with schizophrenia-related social interaction deficits. We were also interested in the associations between measures of emotional contagion and emotional mimicry and individuals with schizophrenia medications and symptoms. To counteract research difficulties due to the variability of fields and methodologies, we considered inclusion and exclusion criteria based on the definition of emotional contagion and emotional mimicry emphasized earlier rather than using specific terminology. 

## 2. Method

This review followed the Preferred Reporting Items for Systematic Reviews and Meta-Analyses (PRISMA 2020 statement). The systematic review was preregistered on PROSPERO (CRD42022383908). In addition, the Covidence website was used to support the different steps of the review. 

### 2.1. Inclusion and Exclusion Criteria

The inclusion criteria were the following: (1) the study must be in English. (2) The study must be cross-sectional or longitudinal. (3) The study must include at least two participant groups, with one group comprising individuals with schizophrenia (ISZ) according to the Diagnostic and Statistical Manual of Mental Disorders third, fourth or fifth edition (DSM-III, DSM-IV, DSM-V), or the Internal Classification of Diseases, 10th edition (ICD-10). The other group serves as a comparison and may consist of individuals with another disorder (e.g., bipolar disorder) or healthy controls (HC). (4) Participants must be 18 years old and older. (5) The study must measure emotional contagion or emotional mimicry. Because some articles reported on emotional contagion and emotional mimicry without explicitly citing these terms, we established the criteria for determining whether the experimental protocol measured these concepts according to their definitions [6,8,11,22] (see Figure 1a,b). To be considered in the review, studies needed propose an experimental procedure with a human or an avatar stimulus expressing emotions through facial expression, vocalization, posture, or movement. Researchers needed to have then measured the participant’s emotional response (for emotional contagion) or the replication of the emotional expression (for emotional mimicry) in reaction to that stimulus. Measuring emotion is possible through the self-reported emotional state, psychophysiological reaction, or brain activity [14]. The measurement of expressions can be conducted by external coders, automated software, or through electromyography of the zygomatic muscles [7]. In addition, we also included studies that measure self-reported susceptibility to emotional contagion via questionnaires such as the Emotional Contagion Scale [23]. (6) Finally, the studies needed to propose inferential statistical analysis to illustrate the differences or lack of differences between groups. 

The exclusion criteria were the following: -Concerning the emotional stimulus: (1) Studies not using emotional expressions as stimuli (e.g., the International Affective Picture System, IAPS [24]). (2) Studies that used oral presentations of semantically emotional words without specifying an emotional intonation in the voice. (3) Studies that used a stimulus in which the emotion component acts as a distraction (e.g., for a memory task) or incongruent emotional stimuli (e.g., happy face with crying sounds).-Concerning the task: (4) Emotional processing studies (i.e., brain activation), in which the participants must complete a task while viewing the emotional stimulus other than rating their emotional experience (e.g., gender discrimination, age discrimination). We chose to exclude these studies, as neural responses are modulated by task instructions [25]. (5) Studies using passive oddball paradigms, as neurological measures can be modulated by the novelty of the deviant stimulus rather than its significance [26]. (6) Studies about imitation, in which researchers explicitly asked the participants to imitate or copy the emotional expression.

### 2.2. Search Strategy

A systematic search was conducted on three electronic databases: PubMed, Web of Science, and PsycINFO. The combination of keywords was the following: (“schiz*” or “psychosis”) and (“mood” or “emotion*” or “affect*”) and (“convergence” or “contagio*” or “transfer” or “social induction” or “synchrony” or “sharing” or “empathy” or “resonance” or “mimicry” or “imitation” or “processing”). We applied two filters: the search keywords must appear in the abstract or title, and only studies published in English were included. Additional hand searches were conducted on Google Scholar, in addition to backward and forward citations. The final search was conducted in April 2023. 

### 2.3. Filtering of Documents

All studies were then transferred to Covidence, where the duplicates were removed. During the selection phase, three independent raters reviewed the titles and abstracts of each study based on the inclusion and exclusion criteria. Disagreements were solved by seeking consensus. Two independent researchers reviewed the full texts of the selected studies to ensure they met all the specified criteria. Disagreements were solved by seeking consensus.

### 2.4. Data Extraction

For selected studies, two independent researchers collected data. The primary outcome sought was the measurement of emotional contagion and/or emotional mimicry in individuals with schizophrenia compared to other subjects. The additional outcomes sought were the relationships between the main outcome and the symptomatology and medication for ISZ. An extraction grid was created on Covidence with the following items: (1) participants’ demographics, (2) sample characteristics, (3) study methodology, and (4) results of main and additional outcomes.

### 2.5. Risk of Bias

The Newcastle–Ottawa Quality Assessment Tool [27] for case–control studies was applied to assess the risk of bias. Studies were evaluated on the selection process (definition and representativeness of cases and definition and selection of controls), the comparability between cases and controls (depending on group matching), and the exposure procedure (ascertainment of exposure, same method for both groups, and no-response rate). Studies could receive a maximum of 10 points, distributed as follows: 4 points for selection, 2 points for compatibility, 1 point for exposure, and an additional point if they included an a priori statistical power analysis.

### 2.6. Synthesis

A narrative synthesis was conducted by grouping articles depending on the outcome of interest (i.e., emotional contagion or emotional mimicry). Articles measuring emotional contagion were subdivided according to their methodology (brain activity, psychophysiological reactions, self-reported emotion, self-reported susceptibility). Additional outcomes (i.e., link to symptomatology and medication) were reported separately. All studies were included in the narrative synthesis. 

## 3. Results

In total, 14,474 records were identified across the three databases. After 5825 duplicates were removed, 8649 titles and abstracts were screened. Overall, 8364 studies were evaluated as irrelevant to the subject. In total, 385 full texts were assessed for eligibility. After this stage, 343 studies were excluded for several reasons: wrong outcome (e.g., studies about emotional recognition or memory or attention), wrong emotional stimuli (e.g., studies without human or avatar emotional expressions), studies using active tasks associated with measures of brain activity (e.g., gender discrimination), wrong sample (e.g., no schizophrenia group or no control group), missing statistical analysis (e.g., no inferential statistics for the main outcome), wrong study design (e.g., literature review), and insufficient information (e.g., no description of the emotional stimuli). Finally, 42 studies respected the inclusion criteria and were included in the review (see Figure 2). 

### 3.1. Risk of Bias

The results of the risk of bias assessment are shown in Table 1. All studies acquired acceptable risk of bias scores. Thus, no studies were excluded based on quality. However, only one study conducted an a priori power analysis. Therefore, it is possible that effects were not detected due to limited statistical power.

### 3.2. Emotional Mimicry

Six studies were included for emotional mimicry. In total, 156 ISZ were compared to 157 HC in measures of facial emotional mimicry via electromyography (EMG) [58,61,64,65,66] and outside coders of facial expression [38]. Mimicry was induced through various methods: using pictures or short video clips of facial expressions [38,65,66], a combination of vocalization and facial expression [61], an interactional paradigm, in which the participants had to narrate an emotional memory [58], and an avatar [64]. In each study, emotional mimicry was measured in reaction to both positive and negative emotional stimuli (see Table 2 for descriptions of each study). 

In studies of emotional mimicry, electromyography (EMG) is employed to assess activity in the zygomaticus major (the muscle responsible for smiling) and the corrugator supercilii (the muscle used for frowning). It is anticipated that positive stimuli will elicit increased activity in the zygomaticus major, while negative stimuli will provoke heightened activity in the corrugator supercilii. Using EMG, two studies found no differences in facial muscle activity between ISZ and HC and demonstrated that both groups displayed the expected facial reaction [58,64]. In contrast, three studies showed that ISZ do not always produce the expected mimicry reactions, in contrast to HC [61,65,66]. According to these studies, ISZ exhibited either non-valence-specific facial reactions to stimuli [61] or decreased facial reactions compared to HC [65,66]. Emotional mimicry impairments were also identified in a study that utilized external coders to analyze facial expressions. Haker and Rossler (2009) observed fewer instances of yawning/sighing or laughing/smiling in response to short video clips among ISZ compared to HC. Additionally, the study noted a significantly higher frequency of incongruent reactions in ISZ compared to HC [38].

### 3.3. Emotional Contagion through Brain Activity

Twenty-two studies were included for measuring emotional contagion through brain activity. In total, 560 ISZ were compared to 594 HC, 72 individuals with major depression disorder (MDD) and bipolar disorder (BD), 40 relatives, and 42 individuals with autism spectrum disorder (ASD). Three methodologies were employed to measure brain activity in response to emotional stimuli: functional magnetic resonance imaging (fMRI) [31,32,33,35,37,39,40,42,44,49,50,53,54,56,57,60,62], electroencephalography (EEG) [30,43,45,69], and magnetoencephalography (MEG) [55]. The emotional stimuli were either pictures of facial expressions [30,31,32,37,39,40,43,45,57,60,62,69], short video clips of facial expressions [35,42,44,54,55,56], or emotional vocalization [33,53]. Most of the studies assessed emotional contagion for positive and negative emotions, with six studies focusing exclusively on negative emotions [31,42,43,49,50,54]. 

Seven studies measured the amygdala activation in response to emotional stimuli, as the amygdala is considered a key region in emotional processing [70]. The results were quite dispersed, as some studies reported that ISZ showed hypoactivation [31,37,60], while others suggested hyperactivation [33,39,50], and one study found no between-group differences for the amygdala activity [49] in response to emotional stimuli. 

Four studies measured the activation of the insula, another key brain region involved in emotional processing [70]. As for measures of the amygdala, the results were dispersed: two studies found hypoactivation of the insula in ISZ compared to HC [35,49], one study found hyperactivation of the insula in ISZ compared to HC [53], and one study found no between-group differences [42]. 

Ten studies measured activation in several other brain regions than the amygdala and the insula. Whilst divergent results were found, most of the studies demonstrated significant differences in brain activity between ISZ and HC [32,37,39,45,53,54,55,57,62,69]. In addition, two studies extended these findings to non-affected relatives of ISZ, highlighting a potential genetic liability [37,62]. In contrast, few studies showed no differences in brain region activity between ISZ, HC, and other primary affective disorders [30,40,42,56]. 

Finally, five studies measured the connectivity throughout the brain regions in response to emotional stimuli in ISZ and HC. The results showed altered connectivity in ISZ compared to HC using both fMRI [31,32,33,44] and MEG [55]. 

### 3.4. Emotional Contagion through Psychophysiological Reactions 

Three studies were included for measuring emotional contagion via psychophysiological reactions (other than the neurological reactions), comparing 66 ISZ to 47 HC. The galvanic skin response [28,67,68], breathing rate, heart rate, and skin temperature were used as markers of psychophysiological reactions. Emotional contagion was induced by presenting pictures of facial expressions [67,68] or a virtual reality head mount with an avatar narrating an emotional story [28]. Two studies were solely interested in negative emotions [67,68], while one study measured physiological reactions to both positive and negative emotions [28].

Two of the three studies in the latter section show similar results, yielding the conclusion that ISZ displayed stronger physiological reactions than HC. The authors found that participants with paranoid schizophrenia displayed more skin conductance in response to negative emotional stimuli than participants with nonparanoid schizophrenia, among whom the responses were already higher than in HC. The third study, extended these results to positive emotional stimuli, showing that ISZ displayed differentiated physiological responses, namely heart rate, skin conductance, and skin temperature, compared with HC [28]. 

### 3.5. Emotional Contagion through Self-Reported Emotion 

Eleven studies were included for measuring emotional contagion through self-reported emotion (six of these studies were also included in the neurological studies section). A total of 308 ISZ were compared to 250 HC across six methods to assess participants’ self-reported emotional states: the Self-Assessment Manikin (SAM) [32,71], the Positive and Negative Affect Schedule (PANAS) [36,37,57,59,60,72], the Emotional Self-Rating scale (ESR) [57,60,73], a scale for emotion [42,44,46,56,59], multiple-choice questions [51], and an implicit measure of emotion [63], based on the Affect Misattribution Procedure [74]. The emotional stimuli were either pictures of facial expressions [32,36,37,57,59,60,63,75] or short clips of an actor displaying emotional expressions [42,44,46,56]. All the studies induced emotional contagion using both positive and negative emotional stimuli, except for one study that specifically focused on pain [42]. 

Most of the studies yielded the same conclusion, demonstrating no significant difference between ISZ and HC. In other words, ISZ and HC were both subject to emotional contagion from positive and negative emotions according to their self-reported emotions [32,37,42,44,51,56,57,59,60,63]. However, despite finding effective emotional contagion for ISZ and HC, some studies reported less or more intense emotional contagion for ISZ compared to HC [36,46,75]. Finally, one study found that anhedonic ISZ were not able to experience the emotional contagion of sadness compared to non-anhedonic ISZ and HC [63]. 

### 3.6. Emotional Contagion through Self-Reported Susceptibility

Six studies were included for measuring self-reported susceptibility to emotional contagion using the Emotional Contagion Scale [23] or the Emotion Contagion subscale of the Questionnaire of Cognitive and Affective Empathy [75]. In total, 462 ISZ were compared to 293 HC. One study also compared ISZ and HC with BD (n = 213) and MDD (n = 163) [48]. 

The Emotional Contagion Scale is a 15-item measure of the tendency to converge emotionally with other individuals. It reports overall susceptibility to emotional contagion and a specific score for five emotions: love, happiness, anger, fear, and sadness [23]. A higher score indicates that the participant is more subject to experience emotional contagion. For the overall score of susceptibility, one study found no difference between groups [34], while another one reported a higher overall score for ISZ compared to HC [47]. For the negative emotions score, all the studies found that ISZ reported being more prone to experience emotional contagion of negative emotions compared to HC [29,34,47]. For the positive emotions score, one study found that ISZ reported a diminished susceptibility to love compared to HC. However, two studiesfound no differences between the groups for the susceptibility to positive emotional contagion. 

The Questionnaire of Cognitive and Affective Empathy is composed of 31 items and five subscales (perspective taking, online simulation, emotion contagion, proximal responsivity, and peripheral responsivity) [75]. For the emotion contagion subscale, two studies reported similar results, showing significantly higher scores for ISZ compared to HC. Despite reporting similar means and large effect sizes, these results were not corroborated by another studywho found no significant differences between ISZ and HC. However, they found that ISZ showed less susceptibility to emotional contagion than MDD [48].

### 3.7. Associations with Symptomatology and Medications 

To gain a comprehensive understanding of emotional contagion and emotional mimicry alterations in ISZ, it is essential to explore their relationships with symptomatology and medication. Among the forty-two studies reviewed, twenty-three investigated associations between measures of emotional contagion or emotional mimicry and symptoms of ISZ (n = 19) and medication for ISZ (n = 15). Overall, only one study reported a significant correlation with medication for ISZ: according to the authors, incongruent mimicry is negatively correlated with the dosage of antipsychotics [38]. Regarding symptoms of ISZ, while nine studies found no significant correlations, ten studies reported significant associations. Specifically, emotional mimicry showed a consistent negative correlation with negative symptoms [38,58,66]. Additionally, emotional contagion measured through brain activity was associated with both positive and negative symptoms [50,57,60,69]. Psychophysiological reactions linked to emotional contagion showed positive correlations with symptoms of delusion, suspiciousness, and persecution [68]. Self-reported emotional contagion indicated both positive and negative correlations with symptoms such as hallucination, delusion, and anhedonia [59,60,63]. Interestingly, all three studies exploring associations between self-reported susceptibility to emotional contagion and symptoms did not find any significant correlations [41,48,52]. 

## 4. Discussion

To the best of our knowledge, this review represents the first comprehensive and systematic analysis of the available data on emotional contagion and emotional mimicry in ISZ. To perform the review, we used the recent definitions of both of the following terms: emotional contagion, as the transfer of an emotional state between individuals resulting in emotional convergence; and emotional mimicry, as the imitation of emotional expressions resulting in matching behaviors.

Our review has three major findings. First, ISZ appear to show emotional mimicry impairments. Second, emotional contagion measures through brain activity and psychophysiological activity are altered in ISZ, whereas self-reported measures seemed to indicate no difference compared to HC. Finally, emotional mimicry and emotional contagion impairments are associated with positive and negative symptoms of schizophrenia, but not with antipsychotic medication. In the following sections, first, we discuss the results, and then we present the clinical implications, the limitations of the literature included in the review, guidelines for future studies, and the limitations of our review. 

### 4.1. Emotional Mimicry

The findings revealed abnormalities in emotional mimicry among ISZ; specifically, ISZ exhibited reduced or inappropriate emotional mimicry compared to HC. Previous research has also documented that ISZ demonstrate fewer emotional expressions during social interactions compared to controls [4,76,77]. While the general decrease in facial expressivity in ISZ may contribute to reduced emotional mimicry, only one study reported a negative correlation between reduced facial expressivity in individuals with schizophrenia and their ability to mimic emotions [58]. Furthermore, one study included in the review compared facial reactions to human faces (i.e., emotional mimicry) and to IAPS stimuli (e.g., car accident) [66]. According to Varcin et al., alterations in facial expressivity may be specific to emotional mimicry, as individuals with schizophrenia (ISZ) displayed expected reactions to IAPS stimuli but not to human facial expressions. The observed expression alterations in response to human faces could be attributed to the negative symptoms of ISZ, which are linked to reduced associability (e.g., social withdrawal, and diminished interest in social interaction). Emotional mimicry relies on the inclination for affiliation [11], and diminished social interaction desire in ISZ could lead to decreased or inappropriate emotional mimicry. This hypothesis finds partial support in the fact that half of the emotional mimicry studies reported negative correlations with the negative symptoms of ISZ [38,58,66]. Finally, medications used to treat schizophrenia could affect facial expressivity because they can inhibit muscle activity. However, in this review, only one study reported an association between measures of emotional mimicry and medications [38].

### 4.2. Emotional Contagion through Brain Activity 

In accordance with recent meta-analyses on emotional processing in schizophrenia [78], the studies included in this review reported altered activation and connectivity in several brain regions during emotional contagion in this clinical population. However, we must also point out that variations and inconsistencies were found across the studies. For instance, both hyper- and hypoactivation of the amygdala were reported in ISZ compared to HC. The different methodologies could account for these discrepancies. However, some studies used very similar experimental procedures and still found opposite results [50,68]. Nevertheless, these inconsistencies are not entirely surprising, given that one of the leading theories of schizophrenia suggests that ISZ may exhibit both an excess of subcortical dopamine and a deficiency of prefrontal dopamine [79]. In line with this theory, the assumption was made that the random firing of dopaminergic neurons could result in the incorrect assignment of significance to neutral objects or situations [80,81,82]. Indeed, ISZ are more prone to perceiving negative emotions in neutral stimuli due to chaotic dopamine transmission [82]. For instance, it was demonstrated that ISZ showed increased activation in regions such as the amygdala or the insula in response to emotionally neutral faces [83]. Therefore, another difficulty in interpreting the results of the studies included in this review is that altered brain activity in ISZ might not be specific to emotional stimuli; thus, they might not reveal specific emotional contagion impairments. In line with this hypothesis, Belge and colleagues (2017) supported the idea that ISZ showed a generalized deficit in face processing rather than a specific emotional processing deficit [84]. 

Consequently, despite the many brain activation and connectivity alterations reported in this review, it is difficult to infer a conclusion on susceptibility to emotional contagion in ISZ based on brain activity studies.

### 4.3. Emotional Contagion through Psychophysiological Activity

Studies measuring psychophysiological reactions such as skin conductance reported increased activation in ISZ compared to HC. As for emotional mimicry, the increased physiological reaction could be specific to human emotional expressions. Indeed, other studies did not report significant physiological differences between ISZ and HC in response to emotional scenes [85,86]. Williams and colleagues (2004, 2007) hypothesized that increased physiological reactions might be linked to positive symptoms, such as paranoid delusions, as they were solely interested in reactions to negative emotions. This hypothesis is reinforced by the findings that individuals with paranoid schizophrenia showed a greater physiological increase compared to non-paranoid schizophrenia [67,68]. In addition, one study reported that increased physiological reactions are correlated with levels of suspiciousness and delusions among individuals with schizophrenia [68]. 

### 4.4. Emotional Contagion through Self-Reported Emotional State and Self-Reported Susceptibility

In contrast to neurological and physiological measures of emotional contagion, no significant differences were found between ISZ and HC in terms of the self-reported emotional states in reaction to emotional stimuli. In addition, either similar or higher levels of self-reported susceptibility to emotional contagion were found for ISZ compared to HC. Hence, the explicit measures of emotional contagion appear to show no alterations of emotional contagion in ISZ. This is corroborated by a meta-analysis that found no differences between ISZ and HC in their subjective hedonic reactions to any emotional stimuli [87]. It is worth noting that self-reported measures are considered valid in schizophrenia, as they were shown to be stable across time, despite symptom and medication evolutions [88]. 

### 4.5. Discordance between Measures of Emotional Contagion 

Importantly, our systematic review highlighted a discordance between brain or psychophysiological activity and self-reported measures of emotional contagion. While the brain and psychophysiological measures demonstrated an altered experience of emotional contagion, the self-reported measures showed no differences between groups. As highlighted in the introduction, the key variable in emotional contagion is the subjective feeling state, which can be assessed through either self-reported emotions or physiological reactions occurring during emotional experiences [89]. A significant distinction between these measurements of emotions lies in the level of control exerted by participants; self-reported emotions are more deliberate, demanding greater attentional resources and introspection compared to physiological reactions [90]. Regarding emotional contagion in ISZ, previous studies have shown that ISZ exhibit reduced visual attention to emotional stimuli compared to HC [91,92]. Consequently, one explanation could be that self-reported measures of emotion, requiring increased visual attention to the stimulus, reduced the attentional gap between ISZ and HC. Thus, while no difference between groups was found for self-reported emotion, measures of brain and psychophysiological activity during passive viewing tasks may still be influenced by ISZ exhibiting reduced visual attention to emotional stimuli. This hypothesis is supported by the fact that increased attentiveness to emotional expression is supposed to improve susceptibility to emotional contagion [8]. 

### 4.6. Clinical Implications

One might question the implication of altered emotional contagion and emotional mimicry in ISZ. Emotional mimicry serves as a social regulator, primarily functioning to “smooth social interactions and establish or maintain social bonds” [11]. Displaying congruent emotional expressions during interactions demonstrates our understanding of the other person’s emotions and our alignment with their perspective, thereby enhancing mutual liking and the quality of social interactions [12,13,93]. In contrast, displaying reduced or inappropriate emotional mimicry is detrimental to social interactions [11,12]. 

Regarding emotional contagion, our study underscores diverse experiences of emotional contagion among ISZ, which may significantly impact their social functioning. Indeed, emotional contagion is an “iterative process that brings individuals closer” [7]. It is associated with social interaction quality, promoting affiliation, affective bonding, joint attention, and empathy [94]. In contrast, counter-contagion (i.e., feeling an incongruent emotional state from the interacting partner) increases social distance and worsens the interaction [7]. 

From a clinical approach, these results open new perspectives for the social rehabilitation of ISZ. Indeed, emotional mimicry can be improved by reinforcing its antecedent: the affiliation stance toward others [11]. In addition, in Hatfield’s definition of emotional contagion (which also includes emotional mimicry), the author emphasized characteristics that make an individual more susceptible to emotional contagion such as increased attentiveness to others and their emotional expressions. Because attentiveness to others’ emotional expression is crucial, decreased visual attention towards emotional features among ISZ should be emphasized to improve emotional contagion and emotional mimicry. Finally, it would be valuable to explore whether social cognition training could enhance emotional contagion and emotional mimicry, given their demonstrated positive effects on cognitive and affective processes [95]. 

### 4.7. Limitations of the Existing Research

There are limitations within the literature reviewed, underscoring the necessity for future studies. The methodological variances between the studies stand out as a significant limitation, as these choices probably contributed to the discrepancies observed across the studies. Due to the impracticality of documenting every single methodological difference, we will only provide a few examples to demonstrate our point. For example, the two studies that did not find emotional mimicry impairments between ISZ and HC diverged significantly in methodology from the other studies. Indeed, one study employed an interactional paradigm, while the other differed by not inducing mimicry with a human stimulus. Additionally, these studies also differed in their choice of data analysis. Indeed, most studies on mimicry only measure the emotional expression of the participant in reaction to standardized stimuli; however, Riehle and Lincoln’s study analyzed mimicry by computing a cross-correlational analysis of both the interacting partner and the participant. The same conclusions can be drawn for studies interested in the measurement of brain activity. In these studies, the methodologies differed in terms of the stimulus presentation (duration, type of stimulus, emotional valence), data analysis (regions of interest), and baseline correction (neutral face, fixation cross). Consequently, future studies should follow a standardized methodology to allow comparisons between results. This standardized methodology could incorporate emotional stimuli specifically designed to assess both emotional contagion and emotional mimicry [96]. In addition, following recent guidelines, the assessment of emotional contagion through self-reported emotions should include pre- and post-induction measurements [7]. 

Another limitation of the literature concerns the association between emotional contagion and emotional mimicry and medication for ISZ. Indeed, less than half of the studies included in the review tested correlations with either symptoms or medication, limiting the interpretation of results. As highlighted in the previous section, we believe that negative symptoms of schizophrenia, such as blunted affect and reduced sociability, could account for deficits in emotional mimicry. Conversely, positive symptoms, including delusions and hallucinations, may explain impairments in emotional contagion, as measured by psychophysiological and brain activity indicators. To corroborate these hypotheses, it is crucial that studies also test the association between both emotional contagion and emotional mimicry and more specific measures of positive and negative symptoms, such as the Revised Paranoid Thoughts Scale (R-GPTS) [97] and the Clinical Assessment Interview for Negative Symptoms (CAINS) [98].

### 4.8. Future Studies 

Our systematic review highlights the need for future studies to continue assessing emotional contagion and emotional mimicry in ISZ. In addition, future studies could also include a multimodal evaluation of emotional contagion to compare explicit and implicit measures. It would also be interesting to compare self-reported susceptibility to emotional contagion and experimentally measured emotional contagion. Indeed, assessing emotional contagion among the same participants through different methodologies would allow a better understanding of the differences in the experience of emotional contagion between ISZ and HC. 

Furthermore, to our knowledge, the literature still lacks clear guidelines for measuring emotional mimicry and emotional contagion in interactive settings. When assessing mimicry, it is crucial to control the timing of the emotional expression of the stimulus to ensure that participants’ facial responses are indeed reactions to the emotional cues presented. However, current analyses of synchrony, such as cross-correlation, lack precision because they calculate an average synchronization score for the entire interaction rather than analyzing specific responses to emotional expressions. As a result, researchers should prioritize developing new analytical methods for social interactions. One potential approach could involve first detecting facial expressions in both the sender and the receiver using external judges, followed by quantifying mimicry in the receiver through measures of cross-correlation. Despite these challenges, accurately measuring emotional mimicry during social interactions remains critical in schizophrenia research, as it aims to address social deficits in individuals with schizophrenia. New methodologies would enable the exploration of the effects of impairments in emotional contagion and mimicry on the quality of social interactions in individuals with schizophrenia in more naturalistic settings.

### 4.9. Limitation of Our Review 

We acknowledge that our systematic review has certain limitations. Firstly, due to the lack of consistency in terminology, it was challenging to ensure that all the relevant articles were included. Nevertheless, we employed a comprehensive search strategy, considering every relevant keyword in our database searches to maximize the number of articles captured. Secondly, the ambiguity in the terminology necessitated the establishment of specific inclusion criteria to categorize studies as investigating either emotional contagion or emotional mimicry. We recognize that some may disagree with our criteria; however, the literature still lacks a definitive methodological framework for defining emotional contagion. Therefore, we proposed a clear set of criteria to address this gap.

### 4.10. Conclusions

To conclude, our review identified a wide range of studies on emotional mimicry and emotional contagion in schizophrenia. The results suggest that individuals with schizophrenia exhibit alterations in both emotional mimicry and contagion compared to healthy controls. However, more research, using standardized methodologies, is needed to replicate and validate these findings. Future studies should also examine the impact of impaired emotional contagion and mimicry on social functioning in this clinical population. Advancing this area of research would benefit from the development of methodologies that facilitate the measurement of emotional mimicry and contagion in interactive settings. Furthermore, we believe that evaluating and addressing these impairments should be integral components of future social cognition interventions for individuals with a diagnosis of schizophrenia.

## Figures and Tables

**Figure 1 jcm-13-05296-f001:**
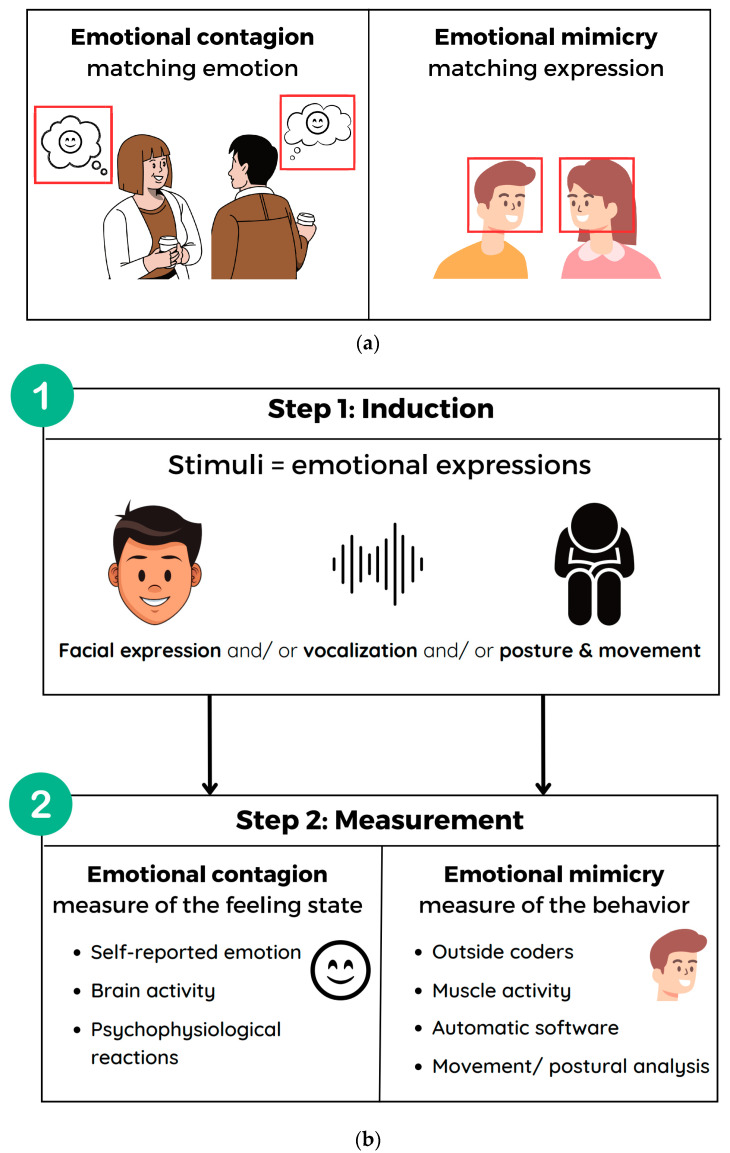
(**a**) Variables of interest in emotional contagion and emotional mimicry. (**b**) Experimental procedures for measuring emotional contagion and emotional mimicry.

**Figure 2 jcm-13-05296-f002:**
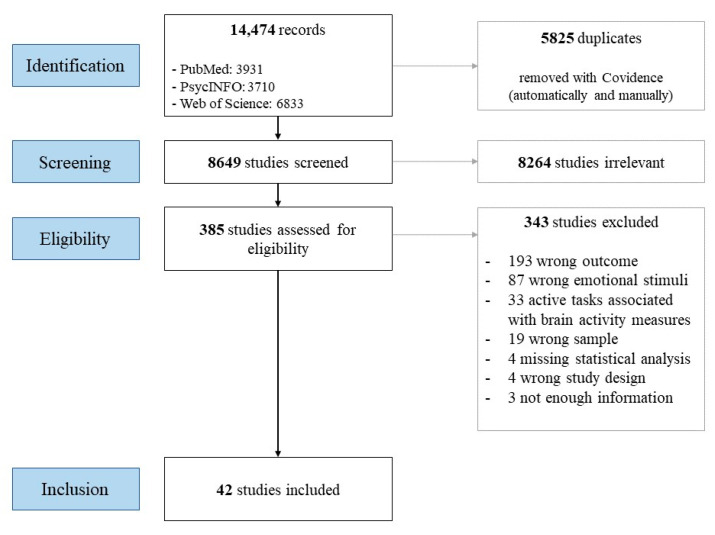
Flowchart of systematic literature review.

**Table 1 jcm-13-05296-t001:** Assessment of studies risk of bias using the The Newcastle–Ottawa Quality Assessment Tool.

Study	Selection	Comparability	Exposure	Power-Analysis
Bekele et al., 2017 [28]	◊◊◊	◊◊	◊◊◊	
Berger et al., 2019 [29]	◊◊◊	◊◊	◊◊◊	
Culbreth et al., 2018 [30]	◊◊◊◊	◊◊	◊◊	◊
Das et al., 2017 [31]	◊◊◊	◊◊	◊◊◊	
Dyck et al., 2014 [32]	◊◊◊	◊◊	◊	
Escarti et al., 2010 [33]	◊◊	◊	◊◊	
Falkenberg et al., 2008 [34]	◊◊◊◊	◊◊	◊◊◊	
Ferri et al., 2014 [35]	◊◊◊	◊◊	◊◊	
Habel et al., 2000 [36]	◊◊◊	◊	◊◊	
Habel et al., 2004 [37]	◊◊	◊◊	◊	
Haker and Rossler (2009) [38]	◊◊◊	◊◊	◊◊	
Holt et al., 2006 [39]	◊◊◊	◊◊	◊◊◊	
Horan et al., 2014 [40]	◊◊◊◊	◊◊	◊◊	
Horan et al., 2015 [41]	◊◊◊◊	◊◊	◊◊◊	
Horan et al., 2016 [42]	◊◊◊◊	◊◊	◊◊	
Horley et al., 2001 [43]	◊◊◊	◊◊	◊	
Hyatt et al., 2022 [44]	◊◊◊◊		◊◊	
Jetha et al., 2013 [45]	◊◊◊◊	◊◊	◊◊◊	
Koevoets et al., 2022 [46]	◊◊◊◊	◊◊	◊◊	
Lehmann et al., 2014 [47]	◊◊◊◊	◊◊	◊◊◊	
Liang et al., 2020 [48]	◊◊◊◊	◊	◊◊◊	
Lindner et al., 2014 [49]	◊◊◊	◊◊	◊◊	
Lindner et al., 2016 [50]	◊◊◊	◊◊	◊	
Mathews and Bach, 2010 [51]	◊◊◊◊	◊◊	◊◊	
Michaels et al., 2014 [52]	◊◊◊◊	◊◊	◊◊◊	
Mitchell et al., 2004 [53]	◊◊◊	◊	◊◊	
Mothersill et al., 2014 [54]	◊◊	◊◊	◊◊	
Popov et al., 2014 [55]	◊◊◊	◊◊	◊◊◊	
Regenbogen et al., 2015 [56]	◊◊	◊	◊◊◊	
Reske et al., 2007 [57]	◊◊◊◊	◊◊	◊◊	
Riehle and Lincoln, 2018 [58]	◊◊◊◊	◊◊	◊	
Schneider et al., 1995 [59]	◊◊◊	◊	◊◊	
Schneider et al., 1998 [60]	◊◊	◊◊	◊◊	
Sestito et al., 2013 [61]	◊◊◊◊	◊◊	◊◊◊	
Spilka et al., 2015 [62]	◊◊◊◊	◊◊	◊	
Suslow et al., 2003 [63]	◊◊◊◊	◊◊	◊◊◊	
Torregrossa et al., 2019 [64]	◊◊◊◊	◊◊	◊◊◊	
Varcin et al., 2010 [65]	◊◊◊◊	◊◊	◊	
Varcin et al., 2019 [66]	◊◊◊◊	◊◊	◊	
Williams et al., 2004 [67]	◊◊◊	◊◊	◊◊◊	
Williams et al., 2007 [68]	◊◊◊	◊◊	◊◊	
Williams et al., 2009 [69]	◊◊◊	◊◊	◊◊	

**Table 2 jcm-13-05296-t002:** Summary of the studies included in the review.

Citation	Sample Characteristics	Variable (s) Measured	Measurement Tool	Stimuli	Main Outcomes Measures of Emotional Contagion or Emotional Mimicry	Additional Outcomes Psychotics Symptomatology	Additional Outcomes Medication
Bekele et al., 2016 [28]	ISZ: n = 12 (4W, 8M) mean age: 45.7 (9.4) HC: n = 12 (SD = 6,6) mean age: 44.9 (SD = 9.9) Matched in age and sex.	EC through psychophysiological reactions.	Skin conductance response rate, mean skin conductance level, breathing rate, mean skin temperature.	Avatars in virtual reality settings narrate an emotional memory and produce emotional expressions (enjoyment, surprise, sadness, disgust, anger)	ISZ showed significantly different psychophysiological reactions in response to positive and negative emotions.	Not reported.	Not reported.
Berger et al., 2019 [29]	ISZ: n = 35 (12W, 23M) mean age: 34.84 (SD = 11.0) HC: n = 18 (10W, 8M) mean age: 29.47 (SD = 5.21) Matched in age and sex.	Self-reported susceptibility to EC.	ECS	-	Higher susceptibility to emotional contagion of fear for ISZ compared to HC. No difference between groups for other emotions (happiness, love, fear, and sadness).	Not reported.	Not reported.
Culbreth et al., 2018 [30]	ISZ: n = 37 (16W, 21M) mean age: 44.9 (SD = 7.8) AP (Affective disorder, MDD and BD): n = 37 (15W, 22M) mean age: 44.3 (SD = 9.3) Matched in age and sex.	EC through brain activity.	EEG	Pictures of facial expressions (happy, sad, angry, afraid, and neutral).	No between-group differences for the late positive potential contrast of neutral and emotional expression.	No significant correlation was found.	No significant correlation was found.
Das et al., 2007 [31]	FES: 14 (M) mean age: 20.4 (SD = 3.3) HC: 14 (M) mean age: 23.1 (SD = 5.9) Matched in age and sex.	EC through brain activity.	fMRI	Pictures of fear and neutral facial expressions were presented under conscious (500ms) or unconscious conditions (16 ms of emotional expression and 163 ms of neutral expression).	ISZ showed reduced amygdala activity in response to fearful expressions compared to HC. ISZ showed the reversal of the normal pattern of connectivity between the amygdala and the brainstem, visual cortex, and dorsal and ventral divisions of the medial prefrontal cortex.	Not reported.	No significant correlation was found.
Dyck et al., 2014 [32]	ISZ: 16 (6W, 10M) mean age: 35.94 (SD = 8.98) HC: 16 (6W, 10M) mean age: 34.25 (8.51) Matched in age and sex.	EC through brain activity. EC through self-reported emotion.	fMRI SAM	Pictures of facial expressions (happiness, sadness, neutral).	ISZ showed decreased activation in the left lingual gyrus compared to HC. ISZ showed increased connectivity between early and late processing areas within the visual cortex compared to HC. No between-group differences for self-reported emotional state.	Not reported.	Not reported.
Escarti et al., 2010 [33]	ISZ (H): n = 27 (13W, 14M) mean age: 39.15 (SD = 8.76) ISZ (NH): n = 14 (6W, 8M) mean age: 42.93 (SD = 14.76) HC: 31 (15W, 16M) mean age: 31.34 (10.52)	EC through brain activity.	fMRI	Emotional and neutral words are pronounced in an emotional and neutral tone, respectively.	Different functional connectivity in limbic regions between HC, ISZ (H), and ISZ (NH). ISZ (H) showed increased amygdala and parahippocampal gyrus activation compared to HC and ISZ (NH).	Not reported.	Not reported.
Falkenberg et al., 2008 [34]	ISZ: n = 17 (6W, 11M), mean age: 28.2 (SD = 7.4) HC: n = 17 (6W, 11M), mean age: 27.6 (SD = 5.4) Matched in age and sex.	Self-reported susceptibility to EC.	ECS	-	No difference in the overall score between HC and ISZ. No difference in susceptibility to “joy” and “sadness”. Lower susceptibility to “love” in ISZ compared to HC. Stronger susceptibility to “anger” in ISZ compared to HC.	Not reported.	Not reported.
Ferri et al., 2014 [35]	ISZ: n = 22 (8W, 14M) mean age: 27.45 (SD = 5) HC:22 (10W, 12M) mean age: 28 (SD = 3.77) Matched in age and sex.	EC through brain activity.	fMRI	Video of an actor performing an action (grasping a bottle) with either a neutral, an angry, or a happy face.	ISZ showed decreased activation in the right anterior insula for angry stimulus compared to HC. No between-group differences for the happy stimulus.	No significant correlation was found.	No significant correlation was found.
Habel et al., 2000 [36]	Am ISZ: n = 40 (19W, 21M) mean age: 30.43 (SD = 7.72) AmHC: not specified. mean age: 21.75 (SD = 3.71) Ger ISZ: n = 24 (12W, 12M) GerHC: n = 24 (12W, 12M) mean age: 32.42 (SD = 8.71) In ISZ: n = 29 (male) mean age: 34.69 (SD = 7.41) In HC: n = 29 (19W, 10M) mean age: 28.10 (SD = 1.80)	EC through self-reported emotion.	PANAS	Pictures of facial expressions (sad and happy).	All cultures: ISZ had lower positive and higher negative emotions during happy emotional contagion. Am: no between-group differences. Indian: ISZ showed less positive emotion during happy and sad induction compared to HC. German: ISZ showed more negative emotion following happiness and sadness emotional contagion compared to HC.	Not reported.	Not reported.
Habel et al., 2004 [37]	ISZ: n = 13 (males) mean age: 32.8 (SD = 8.5) Relatives: n = 13 (males) mean age: 33.8 (SD = 8.7) HC: n = 26 (males) mean age: 33.4 (SD = 8.1) Matched in age and sex.	EC through brain activity. EC through self-reported emotion.	fMRI PANAS	Pictures of facial expressions (sad and happy).	For sadness stimuli, ISZ and relatives showed hypoactivation of the amygdala compared to HC. ISZ also showed hypoactivation in other brain regions (left orbitofrontal area, left superior temporal cortex, left precuneus). No between-group differences were found for brain activation following happiness stimuli. Emotional contagion through self-reported emotion was effective for both groups. No significant between-group differences were found.	No significant correlation was found.	No significant correlation was found.
Haker and Rossler 2009 [38]	ISZ: n = 43 (11W, 32M), mean age: 34 (SD = 10) HC: n = 45 (12W, 33M), 35 (SD = 11) Matched in age and sex.	Emotional mimicry.	Judge (clinical psychiatrists) measuring signs of yawning/sighing or laughing/smiling.	Video sequence of 15 s centered on the face laughing, yawning, or neutral.	ISZ showed less mimicry of both laughing and yawning compared to HC. ISZ produced more incongruent reactions than healthy controls.	Mimicry of laughing correlated negatively with the PANSS negative scale (r = −0.348, *p* = 0.02). Incongruent mimicry correlated negatively with the PANSS negative scale (r = −0.408, *p* = 0.007).	Incongruent mimicry correlated negatively with the dosage of antipsychotics (r = −0.33, *p* = 0.014).
Holt et al., 2006 [39]	ISZ: n = 15 (males) mean age: 47.7 (SD = 7.1) HC: n = 16 (males) mean age: 48.2 (SD = 9.6) Matched in age and sex.	EC through brain activity.	fMRI	Pictures of facial expressions (happiness and fear).	ISZ showed increased left hippocampal activation for happy and fearful stimuli compared to HC. ISZ also showed increased right amygdala activation for fearful stimuli compared to HC.	Not reported.	Not reported.
Horan et al., 2014 [40]	ISZ: n = 23 (6W, 17M) mean age: 46.5 (SD = 11.1) HC: n = 23 (7W, 16M) mean age: 46.7 (SD = 6.9) Matched in age and sex.	EC through brain activity.	fMRI	Pictures of facial expressions (happiness, sadness, anger, fear).	No between-group differences. ISZ and HC showed similar brain activation.	Not reported.	Not reported.
Horan et al., 2015 [41]	ISZ: n = 145 (36W, 109M) mean age: 40.9 (SD = 12.4) HC: n = 45 (13W, 32M) mean age: 43.3 (SD = 10.4) Matched in age and sex.	Self-reported susceptibility to EC.	QCAE	-	Higher susceptibility to emotional contagion for ISZ compared to HC.	No significant correlation was found.	Not reported.
Horan et al., 2016 [42]	ISZ: n = 21 (6W, 15M) mean age: 48.2 (SD = 10.4) HC: n = 21 (7W, 14M) mean age: 46.5 (7.1) Matched in age and sex.	EC through brain activity. EC through self-reported emotion.	fMRI Likert scale (from 1 “not painful” to 4 “extremely painful”).	Video of a person listening to a painful sound and showing facial expression from neutral to painful.	No between-group differences. ISZ and HC showed similar brain activation. No between-group differences. ISZ and HC reported similar painful emotions.	Not reported.	Not reported.
Horley et al., 2001 [43]	ISZ: n = 25 (gender not specified) mean age: 33.6 (SD = 7.63) HC: n = 25 (gender not specified) mean age: 34.36 (SD = 9.07) Matched in age and sex.	EC through brain activity.	EEG	Pictures of facial expressions (neutral and angry).	ISZ showed reduced amplitude (P200) and delay latency (N100, P200, N200, P300) compared to HC.	Not reported.	No significant correlation was found.
Hyatt et al., 2022 [44]	ISZ: n = 41 (12W, 29M) mean age: 30.9 (SD = 3.8) HC: n = 55 (27W, 28M) mean age: 29.1 (3.6) ASD: n = 42 (8W, 34M) mean age: 26.8 (SD = 3.6)	EC through brain activity. EC through self-reported emotion.	fMRI Emotional valence scale from 1 to 9.	Video of an actor narrating an emotional (happy, sad, or neutral) personal story displaying nonverbal emotional expressions.	ISZ showed different functional network connectivity state engagement compared to HC and ASD. No between-group difference in EC through self-reported emotions.	No significant correlation was found.	No significant correlation was found.
Jetha et al., 2013 [45]	ISZ: n = 40 (12W, 28M) mean age: 42.2 (SD = 6.4) HC: n = 39 (12W, 27M) mean age: 39.3 (SD = 7.8) Matched in age and sex.	EC through brain activity.	EEG	Pictures of facial expressions (happy, fear, angry, and neutral).	No between-group differences for the P100 amplitude. ISZ showed decreased N170 amplitude compared to HC.	Not reported.	Not reported.
Koevoets et al., 2022 [46]	ISZ: n = 47 (7W, 40M) mean age: 35.88 (SD = 8.24) HC: n = 47 (4W, 43M) mean age: 32.88 (SD = 7.91) Matched in age and sex.	EC through self-reported emotion.	Likert scale from 1 to 7 for positive emotions (compassionate, soft-hearted, warm, tender) and negative emotions (worried, distressed, disturbed, upset, troubled, and agitated).	Pictures of facial expressions followed by short clips (10 s) of the same person expressing the same emotion.	ISZ reported higher positive and negative emotions compared to HC.	Not reported.	Not reported.
Lehmann et al., 2014 [47]	ISZ: n = 55 (23W, 32M) mean age: 39.8 (SD = 11.9) HC: n = 69 (25W, 30M) mean age: 38.9 (SD = 12.8) Matched in age and sex.	Self-reported susceptibility to EC.	ECS	-	Higher susceptibility in the overall score for ISZ compared to HC. Higher susceptibility to negative emotions (fear, anger, sadness) for ISZ compared to HC. No differences between groups for susceptibility to positive emotions.	Not reported.	Not reported.
Liang et al., 2020 [48]	ISZ: n = 158 (91W, 67M) BD: n = 213 (139W, 74M) MDD: n = 163 (92W, 71M) HC: n = 107 (53W, 54M) ISZ:29.82 (7.1) BD:30.47 (6.28) MDD:30.64 (6.14) HC:29.42 (6.25)	Self-reported susceptibility to EC.	QCAE		Lower emotional contagion scores for ISZ and BD compared to MDD.	No significant correlation was found.	Not reported.
Lindner et al., 2014 [49]	ISZ: n = 36 (14W, 22M) mean age: 30.8 (SD = 7.9) HC: n = 40 (13W, 27M) mean age: 29.5 (SD = 8.3) Matched in age and sex.	EC through brain activity.	fMRI	Picture of facial expressions (disgust, and neutral) presented in conscious (533ms) and nonconscious conditions (33 ms of emotional expression followed by 500 ms of a neutral face).	ISZ showed reduced insula activation compared to HC following masked disgust stimuli. No between-group differences for unmasked stimuli. No between-group differences in amygdala activation.	Not reported.	Not reported.
Lindner et al., 2016 [50]	ISZ: n = 36 (13W, 23M) mean age: 30.6 (SD = 8) HC: n = 42 (13W, 27M) mean age: 29.5 (SD = 8.3) Matched in age and sex.	EC through brain activity.	fMRI	Pictures of facial expressions (fear and neutral) presented in conscious (533 ms) and nonconscious conditions (33 ms of emotional expression followed by 500 ms of a neutral face).	ISZ showed hyperactivation of the amygdala compared to HC. ISZ with affective flattening showed increased amygdala activation compared to HC and ISZ without affective flattening following masked fear stimuli.	Affective flattening is positively correlated with the amygdala response to masked fearful faces (r = 0.52, p < 0.001) and negatively correlated with the amygdala response to unmasked fearful faces (r = −0.4, p < 0.001).	Not reported.
Mathews and Bach, 2010 [51]	ISZ: n = 40 (14W, 26M) mean age: 36.8 (SD = 8.99) HC: 40 (15W, 25M) mean age: 36.30 (SD = 10.47) Matched in age and sex.	EC through self-reported emotion.	Question (“Press 1 if they felt negative, 2 if felt neutral, and 3 if felt positive”)	Pictures of facial expressions (positive, negative, and neutral).	Emotional contagion was effective in both groups. Controls reported experiencing more positive emotions in response to positive stimuli and more negative emotions in response to negative stimuli.	No significant correlation was found.	Not reported.
Michaels et al., 2014 [52]	ISZ: n = 52 (12W, 40M) mean age: 35.3 (SD = 8.8) HC: 37 (17W, 20M), mean age: 33.4 (8.9) Matched in age and sex.	Self-reported susceptibility to EC.	QCAE	-	Higher susceptibility to emotional contagion for ISZ compared to HC.	No significant correlation was found.	No significant correlation was found.
Mitchell et al., 2004 [53]	ISZ: n = 12 (male) mean age: 45.7 (SD = 2.7) HC: n = 13 (male) mean age: 32.2 (SD = 3.6) BD: n = 11 (male) mean age: 42.8 (1.8)	EC through brain activity.	fMRI	An actor reads an emotional scenario (happy, sad, neutral) with the related emotional intonation (happy, sad, neutral).	ISZ showed a reversal of the normal right-lateralized temporal lobe response compared to HC. ISZ showed hyperactivation of the left insula compared to HC.	Not reported.	No significant correlation was found.
Mothersill et al., 2014 [54]	ISZ: n = 25 (5W, 20M) mean age: 42.88 (SD = 10.99) HC: n = 21 (5W, 16M) mean age: 38.24 (SD = 8.62) Matched in age and sex.	EC through brain activity.	fMRI	Short video clips of faces going from a neutral to an angry expression.	ISZ showed weaker deactivation of the medial prefrontal cortex, anterior cingulate cortex, and decreased left cerebellum compared to HC.	Not reported.	Not reported.
Popov et al., 2013 [55]	ISZ: n = 44 (13W, 31M) mean age: 32 (SD = 9.4) HC: n = 44 (20W, 24M) mean age: 29.2 (SD = 7.9) Matched in age and sex.	EC through brain activity.	MEG	Morphed images go from a neutral face to a target facial expression (fearful or happy).	ISZ did not show the sequence of alpha power increase and alpha connectivity decrease compared to HC.	Not reported.	Not reported.
Regenbogen et al., 2015 [56]	ISZ: n = 20 (gender not specified) mean age: 37.3 (SD = 8.44) HC: n = 24 (gender not specified) mean age: 35.25 (SD = 9.8) MDD: n = 24 (gender not specified) mean age: 36.42 (SD = 12.01).	EC through brain activity. EC through self-reported emotion.	fMRI Seven-point scale from “very negative” to “very positive”.	Video clip of an actor telling stories about disgusting, fearful, happy, sad, or neutral situations with either emotional or neutral prosody and facial expression.	No between-group differences in brain activation for the trimodal congruent emotional stimulus. No between-group differences for self-reported emotional state.	Not reported	Not reported.
Reske et al., 2007 [57]	FES: n = 10 (4W, 6M) mean age: 37.4 (SD = 6.06) HC: n = 10 (4W, 6M) mean age: 35.3 (SD = 8.71) Matched in age and sex.	EC through brain activity. EC through self-reported emotion.	fMRI PANAS/ ESR (emotional self-rating scale, unipolar for each emotion).	Pictures of happy and sad facial expressions.	Compared to HC, ISZ showed hypoactivation in the anterior cingulate cortex, orbitofrontal, temporal areas, and hippocampus. No between-group differences for self-reported emotional state. Emotional contagion was effective in both groups according to self-reports.	Therapy and symptom improvement are associated with increased in pre- and postcentral, inferior temporal, and frontal areas for sadness stimuli only.	Not reported.
Riehle and Lincoln, 2018 [58]	ISZ: 28 (16W, 12M) mean age: 41.7 (SD = 10.7) HC: 28 (16W, 12M) mean age 43.0 (SD = 12.1) IP: 28 (16W, 12M) mean age: 39.8 (SD = 13.7) Matched in age and sex.	Emotional mimicry.	EMG	Interacting partners describing emotionally positive and negative memories.	No between-group difference in smiling, smiling mimicry, or frowning.	Smiling activity negatively correlated with the CAINS item reduced facial expressiveness (r = −0.49, p < 0.01) and PANSS N1 blunted affect (r = −0.40, p < 0.05). Frowning activity negatively correlated with the CAINS EXP (r = −0.40, p < 0.05) and the PANSS N1 blunted affect (r = −0.46, p < 0.05). Smiling synchrony negatively correlated with the CAINS reduced facial expressiveness (r = −0.41, p < 0.05).	No significant correlation was found.
Schneider et al., 1995 [59]	ISZ: n = 40 (19W, 21M) mean age: 30.4 (SD = 7.7) HC: n = 40 (not specified) mean age: not specified.	EC through self-reported emotion.	PANAS Unipolar intensity scale from 1 to 5 for happiness and sadness.	Pictures of facial expressions (sad and happy).	No between-group differences: emotional contagion was effective in both groups.	The hallucination subscale of the SAPS was positively correlated with emotional contagion effectiveness (r = 0.40, p < 0.05). Anhedonia was negatively correlated with emotional contagion effectiveness (r not specified).	No significant correlation was found.
Schneider et al., 1998 [60]	ISZ: 13 (males) mean age: 32.46 (SD = 8.03) HC: 13 (males) mean age: 31.69 (SD = 7.65) Matched in age and sex.	EC through brain activity. EC through self-reported emotion.	fMRI PANAS/ ESR (emotional self-rating scale, unipolar for each emotion).	Pictures of facial expressions (sad and happy).	ISZ showed hypoactivation of the amygdala in the sadness induction compared to HC. No between-group differences for self-reported emotional state.	Thought disorder of the SAPS was positively correlated with the activity of the amygdala during happiness contagion (r = 0.58, p < 0.04). Hallucination and delusion subscales of the SAPS negatively were correlated with the emotional contagion of sadness through self-reported emotion (r = −0.56, p < 0.04, r = −0.60, p < 0.03).	No significant correlation was found.
Sestito et al., 2013 [61]	ISZ: 15 (5W, 10M), mean age: 32.8 (SD = 1.7) HC: 15 (5W, 10M), mean age: 35.8 (SD = 2.3) Matched in age and sex.	Emotional mimicry.	EMG	2 s videos of professional actors showing positive, negative, and neutral expressions. Clips included vocalization and facial expressions. Stimuli were presented in different categories (visual, audio, audio–visual congruent).	No difference between groups for the corrugator. HC and ISZ reacted in a similar way to negative emotional stimuli. In the audio-only modality, ISZ showed no activation in reaction to positive stimuli compared to HC. In the audiovisual and visual modalities, ISZ exhibited a nonspecific response (i.e., a similar activation of the zygomaticus for the negative and the positive emotions).	Not reported.	Not reported.
Spilka et al., 2015 [62]	ISZ: n = 28 (13W, 15M) mean age: 41.07 (SD = 11.15) Relatives: n = 27 (17W, 10M) mean age: 41.19 (SD = 15.46) HC: n = 27 (14W, 13M) mean age: 40.7 (SD = 11.1) Matched in age and sex.	EC through brain activity.	fMRI	Pictures of facial expressions (happy, sad, angry, fearful, and neutral).	ISZ and relatives showed hypoactivation in the bilateral FFA (fusiform face area), OFA (occipital face area), and visual cortex in response to sadness stimuli.	Not reported.	Not reported.
Suslow et al., 2003 [63]	Anhedonic ISZ: n = 30 (15W, 15M) mean age: 37.1 (SD = 9.8) Flat affect ISZ: n = 30 (10W, 20M) mean age: 32.9 (SD = 8.4) ISZ: n = 28 (14W, 14M) mean age: 35.7 (SD = 9.4) HC: n = 30 (15W, 15M) mean age: 35.5 (SD = 8.6) Matched in age and sex.	EC through self-reported emotions.	Implicit affective valence measure.	Pictures of facial expressions (happiness, sadness, and neutral).	No between-group differences, except for anhedonic individuals who showed no emotional contagion in response to sadness stimuli.	Anhedonia subscale score of the SANS positively correlated with negative emotional contagion (r = 0.20, p < 0.05) and negatively correlated with positive emotional contagion (r = −0.21, p < 0.05).	No significant correlation was found.
Torregrossa et al., 2019 [64]	ISZ: n = 21 (9W, 12M), mean age: 47.9 (SD = 7.83) HC: n = 23 (13W, 10M), mean age: 45.65 (SD = 8.17) Matched in age and sex.	Emotional mimicry	EMG	Different avatars displayed a neutral face for 2 s and an emotional face (joy, surprise, sadness, fear, disgust, contempt, or anger) for 2.5 s at different intensities (low, medium, high).	No differences between HC and ISZ.	Not reported.	Not reported.
Varcin et al., 2010 [65]	ISZ: n = 25 (15W, 10M), mean age: 42.9 (SD = 9.43) HC: n = 25 (14W, 11M) 39.2 (SD = 10.85) Matched in age and sex.	Emotional mimicry.	EMG	Black and white pictures of facial expressions (angry and happy).	ISZ showed comparable activity with HC for the zygomaticus in response to happy stimuli and for the corrugator in reaction to the angry stimuli. ISZ showed no increased activation of the zygomaticus in the happy compared to the angry stimuli and no increased activation of the corrugator in the angry compared to the happy stimuli, whereas these activations were observed in the HC.	No significant correlation was found.	No significant correlation was found.
Varcin et al., 2019 [66]	ISZ: n = 24 (12W, 12M), mean age: 46.2 (SD = 8.78) HC: n = 21 (13W, 8M), mean age: 44.9 (SD = 13.54) Matched in age and sex.	Emotional mimicry.	EMG	Pictures of happy and angry facial expressions.	ISZ showed less zygomaticus activity while watching the happy stimuli and less corrugator activity while watching the angry stimuli compared to HC.	Zygomaticus activity in response to happy stimuli inversely correlated with levels of negative symptomatology (r = −0.45, *p* = 0.033)	No significant correlation was found.
Williams et al., 2004 [67]	ISZ: n = 27 (10W, 17M) mean age: 27.3 (SD = 9.6) HC: n = 22 (8W, 14M), mean age: 27.2 (SD = 8.1) Matched in age and sex.	EC through psychophysiological reactions.	Skin conductance (number and amplitude).	Pictures of expressions of fear and neutral expression.	ISZ produced more skin conductance (number and amplitude) than HC for fear and neutral expressions. HC showed a significant difference between reactions to neutral and fearful expressions, whereas ISZ did not show a significant difference between fear and neutral stimuli. Paranoid ISZ showed more skin conductance to fear than nonparanoid ISZ (amplitude and number). Paranoid ISZ and nonparanoid ISZ did not differ from neutral.	Not reported.	Not reported.
Williams et al., 2007 [68]	PISZ: n = 13 (5W, 8M), mean age: 26.9 (SD = 9.1) NPISZ: n = 14 (5W, 9M), mean age: 27.8 (SD = 10.4) HC: n = 13 (10W, 17M), mean age: 25.1 (SD = 8.1) Matched in age and sex.	EC through psychophysiological reactions.	Skin conductance (number and amplitude).	Pictures of expressions of fear anger and disgust.	PISZ generated a higher frequency and amplitude of SCRs in response to fear and disgust than HC. PISZ generated more SCRs than HC in response to anger. NPISZ elicited greater amplitude of SCRs than HC for disgust. PISZ elicited greater amplitude of SCRs to fear than NPISZ.	Heightened SCR amplitude correlated with higher levels of suspiciousness/persecution for both fear (r = 0.55, *p* = 0.009) and anger (r = 0.39, *p* = 0.02). A greater number of SCRs to disgust positively correlated with delusions (r = 0.47, *p* = 0.01).	Not reported.
Williams et al., 2009 [69]	FES: n = 28 (8W, 20M) mean age: 19 (SD = 3) HC: n = 72 (18W, 4M) mean age: 20 (SD = 2.8) Matched in age and sex.	EC through brain activity.	EEG	Pictures of facial expressions (fear and happiness) presented in conscious (500 ms) and nonconscious conditions (10ms of emotional expression, followed by 150 ms of a neutral face).	FES showed absolute and relative gamma synchrony alterations under conscious and nonconscious conditions in response to happy and fearful stimuli.	PANSS-negative symptoms were positively predicted by left temporal synchrony (R2 = 0.18).	No significant correlation was found.

Note. Am: American, ASD: Autism Spectrum Disorder, BD: Bipolar Disorder, CAINS: Clinical Assessment Interview for Negative Symptoms, EC: Emotional Contagion, ECS: Emotional Contagion Scale, EEG: Electroencephalography, EMG: Electromyography, FES: First Episode Schizophrenia, fMRI: functional Magnetic Resonance Imaging, Ger: German, HC: Healthy Control, In: Indian, MDD: Major Depressive Disorder, NPISZ: Nonparanoid Individual with Schizophrenia, PANAS: Positive and Negative Affect Scale, PANSS: Positive and Negative Symptom Scale, ISZ: Individuals with Schizophrenia, PISZ: Paranoid Individual with Schizophrenia, ISZ: Individuals with Schizophrenia, QCAE: Questionnaire of Cognitive and Affective Empathy, SAM: Self-Assessment Manikin, SAPS: Scale for the Assessment of Positive Symptoms, SCR: Skin Conductance Response.

## Data Availability

Not applicable.

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
