# Peer review of "Emotional Contagion and Emotional Mimicry in Individuals with Schizophrenia: A Systematic Review"

_jcm, 2024, doi:10.3390/jcm13175296_

Round 1
Reviewer 1 Report
Comments and Suggestions for Authors
It is a good review. this paper is original and I think it will be many readers due to its relevance.
In line 33 put on an example of disorganized symptoms. behavior and speech are not examples. In line 55-59 is confusing the description of emoional mimicry and behabioral mimicry. Moreover, it´s important to clarify the definitions of emotional contagion and emotional mimicry in the way that you consider. You said in lines 60-63 how you consider emotional contagion, but did you take into account this in the selected articles?. May be you only need to clarify the concepts, and you don´t need to discuss this topic in introduction.
In the second parragraph, lines 65-77 you again talk about emotional contagion, emotional mimicry and discuss the definitions. I thin that you can be more specific, or little more pragmatic.
in line 106 you can write down the PROSPERO code, not the link.
I don´t have suggestions about methodology.
it´s clear that you use PRISMA checklist.
Comments on the Quality of English Language
I don´t have comments about quality of English language.
Author Response
Comments 1: It is a good review. this paper is original and I think it will be many readers due to its relevance.
Response 1: Dear Reviewer,
We express our gratitude for dedicating your time and attention to reviewing our manuscript and for providing valuable constructive feedback. Below please find the revised manuscript and our detailed responses to the reviewers’ comments and concerns.
Comments 2: In line 33 put on an example of disorganized symptoms. behavior and speech are not examples. In line 55-59 is confusing the description of emoional mimicry and behabioral mimicry. Moreover, it´s important to clarify the definitions of emotional contagion and emotional mimicry in the way that you consider. You said in lines 60-63 how you consider emotional contagion, but did you take into account this in the selected articles?. May be you only need to clarify the concepts, and you don´t need to discuss this topic in introduction.
In the second parragraph, lines 65-77 you again talk about emotional contagion, emotional mimicry and discuss the definitions. I thin that you can be more specific, or little more pragmatic.
Response 2: Thank you for your suggestions. We clarified and reorganized the introduction in the manuscript. See below:
Hatfield et al., (1994), first defined emotional contagion as the “tendency to automatically mimic and synchronize facial expressions, vocalizations, postures and movements with those of another person and, consequently, converge emotionally” (p. 5). According to this definition, emotional contagion is a three-step process: the first one is the mimicry of an emotional expression, the second one is afferent feedback, and the third one is emotional convergence. This definition of emotional contagion is referred to as primitive emotional contagion due to its description as an automatic and bottom-up process. Two recent literature reviews proposed that emotional contagion can be measured through a participant’s self-reported emotional states, self-reported susceptibility, behavioral expressions, and psychophysiological reactions to another person’s emotional expression 7,14.
However, Hatfield and colleagues’ definition of emotional contagion was contested for different reasons. First, some studies were not able to demonstrate the causal link between mimicry and contagion 15–17. In addition, according to Hess & Fischer (2014), emotional mimicry is not an automatic reaction, rather it carries social functions such as the mimickers showing their comprehension of their counterpart’s emotion. As a result, although certain studies have demonstrated the connection or co-occurrence of emotional contagion and emotional mimicry 18,19, the limitations of Hatfield’s definition have led to the distinction between the two phenomena 6. In addition, in their distinction between emotional contagion and emotional mimicry, Hess & Fischer (2022) note that “emotional contagion refers to a feeling state whereas mimicry refers to an overt behavior” highlighting the different variables of interest of the two phenomena 11.
Consequently, for this review, we selected articles based on the following definitions: Emotional contagion refers to the transfer of an emotional state from one individual to another through emotional expressions, leading to emotional convergence. It is typically measured by participants' self-reported emotions, brain activity, or psychophysiological responses to another individual's emotional expressions, reflecting their own emotional experience. Conversely, emotional mimicry refers to the imitation of emotional expressions, resulting in corresponding behaviors. It is measured by observing the participants' emotional expressions in response to another individual's emotional expression.
Comments 3: in line 106 you can write down the PROSPERO code, not the link.
Response 3: Thank you, we deleted the link and provided the PROSPERO code in the manuscript.
Comments 4: I don´t have suggestions about methodology.
it´s clear that you use PRISMA checklist.
Response 4: Thank you for your feedback.
Reviewer 2 Report
Comments and Suggestions for Authors
This is an interesting study that can be minor revised , and in my personal opinion, Table 2 is too long and it is recommended not to put it in the main manuscript but in the supplementary file.
Comments on the Quality of English LanguageMinor editing of English language required.
Author Response
Comment 1:
This is an interesting study that can be minor revised , and in my personal opinion, Table 2 is too long and it is recommended not to put it in the main manuscript but in the supplementary file.
Response 1:
Dear Reviewer,
We express our gratitude for dedicating your time and attention to reviewing our manuscript. We preferred to keep table 2 in the main manuscript.
Reviewer 3 Report
Comments and Suggestions for Authors
The article deals with a very interesting and little-studied topic about contagion and emotional mimicry in people with psychosis. This is a systematic review.
I have a few minor points:
In the introduction, the symptomatology of schizophrenia also includes cognitive symptoms. The sitntomatology of psychosis should be well described according recent classifications.
Method
I would like the authors to clarify; if only articles that analyse data from people diagnosed with SZ have been included or have also included other psychotic disorders.
I have also noticed that there are some studies that are compared with people with affective disorders, but there is always a control group, except in one of the studies in which only people with psychosis are compared with people with affective disorders (Culbreth et al., 2018), as the authors justify the inclusion of this study?
Results
In the footnote of the table, some abbreviations such as ASD, or BP are missing. Authors should check them.
Discussion
I miss some link between brain activity and psychopathology and, potential perceptual distorssion of emotions.
Author Response
Comments 1: The article deals with a very interesting and little-studied topic about contagion and emotional mimicry in people with psychosis. This is a systematic review.
Response 1: Dear Reviewer,
We express our gratitude for dedicating your time and attention to reviewing our manuscript and for providing valuable constructive feedback. Below please find the revised manuscript and our detailed responses to the reviewers’ comments and concerns.
Comments 2: I have a few minor points:
In the introduction, the symptomatology of schizophrenia also includes cognitive symptoms. The sitntomatology of psychosis should be well described according recent classifications.
Response 2: Thank you for your suggestion. We added the cognitive symptoms in the description of the schizophrenia symptomatology.
Comments 3: Method
I would like the authors to clarify; if only articles that analyse data from people diagnosed with SZ have been included or have also included other psychotic disorders.
I have also noticed that there are some studies that are compared with people with affective disorders, but there is always a control group, except in one of the studies in which only people with psychosis are compared with people with affective disorders (Culbreth et al., 2018), as the authors justify the inclusion of this study?
Response 3: Our main focus was on individuals with a diagnosis of schizophrenia. Consequently, to be included in the review, studies had to include at least one group of individuals diagnosed with schizophrenia. The second criterion was that the study must include a comparison group, which could be either another mental health diagnosis or a group of healthy subjects. We explained this in the method section:
The study must include at least two participant groups, with one group comprising individuals with schizophrenia (ISZ) according to the Diagnostic and Statistical Manual of Mental Disorders third, fourth or fifth edition (DSM-III, DSM-IV, DSM-V), or the Internal Classification of Diseases, 10th edition (ICD-10). The other group serves as a comparison and may consist of individuals with another disorder (e.g., bipolar disorder) or healthy controls (HC).
Comments 4: Results
In the footnote of the table, some abbreviations such as ASD, or BP are missing. Authors should check them.
Response 4: Thank you for your feedback. We corrected it.
Comments 5: Discussion
I miss some link between brain activity and psychopathology and, potential perceptual distorssion of emotions.
Response 5: Thank you for you suggestion, we completed the discussion of our section “Emotional Contagion through Brain Activity”. See below:
Yet, these inconsistencies are not entirely surprising, given that one of the leading theories of schizophrenia suggests that ISZ may exhibit both an excess of subcortical dopamine and a deficiency of prefrontal dopamine 79. In line with this theory, assumptions were made that the random firing of dopaminergic neurons could result in the incorrect assignment of significance to neutral objects or situations 80–82. Indeed, ISZ are more prone to perceiving negative emotions in neutral stimuli due to the chaotic dopamine transmission 82. For instance, it was demonstrated that ISZ showed increased activation in regions such as the amygdala or the insula in response to emotionally neutral faces 83.